# Anti-Inflammatory Effects of Amantadine and Memantine: Possible Therapeutics for the Treatment of Covid-19?

**DOI:** 10.3390/jpm10040217

**Published:** 2020-11-09

**Authors:** Félix Javier Jiménez-Jiménez, Hortensia Alonso-Navarro, Elena García-Martín, José A. G. Agúndez

**Affiliations:** 1Section of Neurology, Hospital Universitario del Sureste, Arganda del Rey, E-28500 Madrid, Spain; hortalon@yahoo.es; 2University Institute of Molecular Pathology Biomarkers, UNEx. ARADyAL Instituto de Salud Carlos III, E-10071 Cáceres, Spain; elenag@unex.es (E.G.-M.); jagundez@unex.es (J.A.G.A.)

**Keywords:** amantadine, memantine, anti-inflammatory effects, SARS-Cov-2, COVID-19, therapy

## Abstract

We have reviewed current data on the anti-inflammatory effects of amantadine and memantine in clinical and in vivo models of inflammation, and we propose that these effects have potential interest for the treatment of the SARS-CoV-2 infection (COVID-19 disease). To that end, we performed a literature search using the PubMed Database from 1966 up to October 31 2020, crossing the terms “amantadine” and “memantine” with “inflammation” and “anti-inflammatory”. Amantadine and/or memantine have shown anti-inflammatory effects in chronic hepatitis C, in neuroinflammation induced by sepsis and by lipopolysaccharides, experimental models of multiple sclerosis, spinal cord injury, and respiratory diseases. Since the inflammatory response is one of the main pathogenetic mechanisms in the progression of the SARS-CoV-2 infection, anti-inflammatory effects of amantadine and memantine could be hypothetically useful in the treatment of this condition. This potential utility deserves further research.

## 1. Introduction

Amantadine (adamantan-1-amine) is an adamantane derivative which was used to treat influenza caused by influenza virus type A (the antiviral effects of amantadine are based on the ability to interfere with the viral M2 protein, therefore interfering with viral replication), although today it is not recommended for this indication because of the development of drug resistance [1,2]. Amantadine is also useful for the treatment of Parkinson’s disease (PD) at early stages (it decreases symptoms of bradykinesia, rigidity, and tremor) and to treat levodopa-induced dyskinesia in this disease. The mechanisms of action of amantadine in PD include non-competitive antagonism of the N-methyl-D-aspartate (NMDA) receptor (NMDAR), an increase of dopamine release, a reduction of dopamine reuptake, and mild anticholinergic (by antagonism of the nicotinic cholinergic receptor) and D2 receptor agonist actions [1,3]. Amantadine is also widely used as an effective and safe drug in the treatment of fatigue in patients with multiple sclerosis [4] and has shown efficacy in persistent vegetative state due to severe traumatic brain injury [5] or to severe cerebral hemorrhage [6].

Memantine (3,5-dimethyladamantan-1-amine) is a non-competitive, low-affinity, voltage-dependent antagonist of NMDA receptors, and also acts as an antagonist of both the nicotinic cholinergic and serotonergic (5-HT) type 3 (5-HT_3_) receptors. Memantine is one of the two classes of drugs approved for clinical use in moderate to severe Alzheimer’s disease (AD) [7,8,9]. Other clinical uses include the treatment of vascular dementia, dementia of Wernicke-Korsakoff syndrome, and acquired pendular nystagmus of multiple sclerosis [10].

Together with these main mechanisms of action and clinical uses, both amantadine and memantine have shown important anti-inflammatory actions. Because inflammatory response with the so-called “cytokine-storm” (triggering viral sepsis and inflammatory-induced lung injury) is one of the main pathogenetic mechanisms in the progression of the SARS-CoV-2 (COVID-19) infection [11], we should speculate on the hypothesis that amantadine and memantine could be useful as adjunctive therapy for the treatment of this condition. For this purpose, we undertook a literature search using the PubMed database from 1966 up to 31 October 2020, crossing the terms “amantadine” and “memantine” with “inflammation” and “anti-inflammatory”. The whole search retrieved a total of 337 references, which were examined manually one by one, before the references strictly related to this issue were selected. The flowchart for the selection of studies is represented in Figure 1.

## 2. Anti-Inflammatory Effects of Amantadine

Amantadine and its derivative rimantadine, in combination with interferon α2b (IFNα2b) and ribavirin, have shown higher efficacy in the treatment of chronic hepatitis C in non-responders to IFNα2b than the combination of IFNα2b with ribavirin, measured as the end-of-treatment response (ETR), according to two literature reviews [12,13].

Amantadine combined with pegylated IFNα2b and ribavirin has also shown efficacy in the treatment of chronic hepatitis C (measured as a sustained virological response—SVR) [14,15], which was higher in drug-naïve than in non-responders to IFNα2b [16].

While several studies have shown the lack of effect of amantadine alone in the treatment of patients with chronic hepatitis C non-responders to IFNα2b, both in reducing viremia and in reducing alanine aminotransferase (ALT) levels [17,18], other studies have shown a sustained reduction in ALT levels in patients treated with this drug, suggesting its potential anti-inflammatory activity [12,19].

Amantadine has shown the ability to reduce the induction of inflammatory factors such as RANTES (regulated on activation, normal T cell expressed and secreted; Chemokine (C-C motif) ligand 5) in a dose-dependent manner, and the activation of p38 MAP (mitogen-activated protein) kinase and c-Jun-NH_2_-terminal kinase (JNK) in cell cultures of bronchial epithelial cell lines infected with influenza virus (the reduction in the activation of p38 MAP kinase and JNK seems to be related to the inhibition of virus replication, because amantadine did not inhibit the activation of these kinases induced by tumour necrosis α -TNF-α) [20].

Both amantadine and memantine showed a positive effect on neurological deficits and in the improvement of rats with experimental allergic encephalomyelitis (EAE), an experimental model of multiple sclerosis, increasing both the number and activation of microglial cells, and decreasing mRNA expression of interleukin-6 (IL-6) and, to a lesser extent, expression of IL-1β, TNF-α mRNA, but did not affect the activation of astrocytes [21].

Amantadine injected intraperitoneally in mice reduces neuroinflammation by attenuation of the increase of IL-1β expression induced in a model of sepsis induced by caecal ligation and puncture [22].

Administration of amantadine to rats with spinal cord injury showed a protective effect related to endothelial development and angiogenesis, reduction in inflammation, and markers of apoptosis and oxidative stress [23].

Finally, amantadine inhibited the release of microglial pro-inflammatory factors and increased the expression of neurotrophic factors such as glial-derived neurotrophic factor (GDNF) from astroglia in rat midbrain mixed neuron-glia cultures challenged with lipopolysaccharide (LPS) or the dopaminergic neurotoxin 1-methyl-phenyl-pyridinium ion (MPP^+^), these actions being independent of NMDA receptor inhibition [24].

## 3. Anti-Inflammatory Effects of Memantine

Memantine has shown the ability to decrease neuroinflammation induced by intraventricular infusion of LPS in rats by reducing microglia activation [25]. Like amantadine, as previously stated, memantine improved EAE in rats [21].

Memantine has shown a potent protective effect against lesions induced by 6-hydroxydopamine (OHDA) in dopaminergic PC12 cells related to reversing nerve growth factor IB (Nurr77) upregulation [26]. Nurr77 activates the nuclear factor-κB (NF-κB) signalling pathway, potentiates the induction of pro-inflammatory gene expression, and enhances mouse resistance to LPS-induced sepsis by inhibiting NF-κB activity and suppressing aberrant cytokine production [26].

Memantine has shown anti-inflammatory effects on human microvascular endothelium induced by the pro-inflammatory cytokine TNF-α in an experimental in vitro blood-brain-barrier (BBB) model of human brain microvascular endothelial cells. Memantine prevents the attachment of monocyte THP-1 cells to human brain microvascular endothelial cells, preventing the increase in expression of cell adhesion molecules, such as intercellular adhesion molecule-1 (ICAM-1), vascular cell adhesion molecule-1 (VCAM-1), and E-selectin, and interfering with monocyte transmigration across the BBB model [27].

Memantine significantly decreased pulmonary inflammation induced by cigarette smoke combined with intratracheal injection of LPS in mice and in cultures of RAW264.7 cells through several mechanisms, which included the attenuation of the increased release of cytokines (TNF-α, IL-6, and IFN-γ) and glutamate induced by cigarette smoke and LPS; and the reduction of NMDA receptor 1 (NMDAR1, NR-1) and the xCT subunit of the x(c)(-) cystine/glutamate antiporter (xCT, a cystine/glutamate exchanger) expression, Ca^2+^ influx, and phosphorylation of extracellular signal-regulated kinase1/2 (ERK1/2) [28].

Memantine reverse prevented the increased expression of NMDAR1, NMDAR2B, Calpain, and Caspase 3 expression, and the decreased level of NMDAR2A, CaMKIIα, and cyclic AMP-response element-binding CREB protein expression; and restored the increased levels of GFAP, TNF-α, and iNOS, and the decreased levels of IL-10 induced by intracerebroventricular injection of streptozotocin (STZ) to rats [29].

## 4. Amantadine, Memantine, and COVID-19

First suggested by Tipton & Wszolek [30], there are many reasons to think that these two non-expensive drugs, amantadine and memantine, could be useful in COVID-19 therapy.

Together with the previously mentioned antiviral effects on amantadine in influenza, based on its ability to interfere with the viral M2 protein, recent reports suggest a possible antiviral action against SARS-CoV-2.

Abreu et al. [31] hypothesized a possible antiviral effect of amantadine by blocking a 5-α-helix channel (“viroporin channel”) in a hydrophobic region of the intramembrane region of COVID-19.

Smieszek et al. [32] showed that amantadine can down-regulate the expression of cathepsin L (CTSL) and other lysosomal enzymes, and these two mechanisms have been proposed as potentially useful in decreasing the ability of SARS-CoV-2 to enter the cells and to decrease virus replication.

This aside, it has been shown that memantine is capable of inhibiting the E protein of SARS-CoV-2, which seems to act as an ion channel, and is related to viral pathogenicity [33]. All these reports focus on the possible antiviral effect of amantadine and memantine (Figure 2).

However, clinical experience regarding the use of amantadine and memantine in the therapy of COVID-19 is limited. In this respect, a recent preliminary report by Rejdak & Grieb [34] has shown that 10 patients diagnosed with multiple sclerosis and five PD patients, all of them under previous treatment with amantadine, along with seven patients with cognitive impairment under previous treatment with memantine (these treatments had been used for at least three months previously to enrolment in the study) with asymptomatic SARS-CoV-2 infection (all these individuals have been in contact with patients diagnosed with COVID-19 and showed positive real-time polymerase chain reaction—RT-PCR), remained asymptomatic after two-week quarantine, suggesting a potential preventive effect of amantadine and memantine against the development of symptomatic COVID-19.

More recently, Aranda-Abreu et al. [35] reported an observational open-label study involving 15 patients infected with SARS-CoV-2 treated with 100 mg/day of amantadine for a period of 14 days. Diagnosis of SARS-CoV-2 infection was done by the presence of clinical symptoms compatible with COVID-19, and confirmation was done by positivity for IgG antibodies and negativity of IgM against SARS-CoV-2 at the end of the treatment. Together with amantadine, 14 patients received azithromycin 500 mg daily for 6 days, aspirin daily for 6 days, and celecoxib 200 mg daily for 6 days; three patients required ipratropium bromide/salbutamol 3 nebulisations per day during 5 days, and 2 oxygen mask 4 Lpm. All the patients recovered successfully.

## 5. Discussion, Conclusions, and Future Directions

Konstantinidou & Papanastasiou [36] recently reviewed the most promising repurposed regimens for the therapy of COVID-19, which included antimalarial (chloroquine and hydroxychloroquine), cardioprotective (colchicines), angiotensin-converting enzyme 2 inhibitors, antiviral (remdesivir, favipiravir, ribavirine, lopinavir/ritonavir), anti-inflammatory (tocilizumab, leronlimab, interferon λ) and antiparasitic drugs (ivermectin, nitazoxanide) among others.

More recently, Sarkar et al. [37] reported the current status and future perspectives of potential therapeutic options for COVID-19, including an important number of drug trials that are currently ongoing. These include antiviral drugs, antimalarial drugs, antibiotics and anti-parasitics, anti-inflammatory and immunosuppressive drugs, kinase inhibitors, monoclonal antibodies, hormonal preparations, and blood and blood-forming organs. According to this review, dexamethasone has been considered as a standard in the treatment of COVID-19, especially in severe cases, since a study conducted in the UK showed a 35% reduction in mortality rate in patients with mechanical ventilation and a 20% reduction in the mortality rate in patients who were given oxygen [38]. In contrast, a recent meta-analysis questioned the efficacy of corticosteroids in COVID-19, concluding that “with the use of corticosteroids, delayed recovery and a longer hospital stay were observed” [39]. Together with dexamethasone, remdesivir has been authorized and recommended for the therapy of COVID-19 in the US, where two studies have shown that this drug accelerated the patient’s recovery compared to placebo, and reduced hospitalization period [37]. Finally, the antiviral Fapipavir has been recommended in China based on the results of two randomized trials [37]. Recombinant monoclonal antibodies, interferon-based therapies, convalescent plasma therapy, small-molecular drug therapies, and cell-based therapies are expected to show promising results while an effective vaccination is being developed.

Despite that to date there are no effective and safe chloroquine or hydroxychloroquine dosage treatments for COVID-19, at least in the more severe (hospitalized) patients [36], and the fact that the combination of these drugs is no longer recommended in the US and by the World Health Organization, a recent retrospective open-label French study involving 1061 patients with confirmed COVID-19 (both symptomatic and asymptomatic, diagnosed through early unrestricted massive PCR screening) treated with the combination of hydroxychloroquine (200 mg three times daily for ten days) plus azithromycin (500 mg on day 1 followed by 250 mg daily for the next four days), has shown excellent results, with good clinical outcome and virological cure in 91.7%, prolonged viral carriage with final relativisation of PCR in 4.4%, and poor clinical outcome in 4.3%, with death in 0.75% of patients (all deaths due to respiratory failure) [40]. Moreover, a double-blind randomized trial to assess the efficacy and safety of hydroxychloroquine chemoprophylaxis in COVID-19 in healthcare personnel in a hospital setting is undergoing in Spain [41].

The possible usefulness of amantadine and memantine in the treatment of COVID-19 lies in two pharmacological effects: antiviral action and anti-inflammatory effects. The anti-inflammatory effects are summarized in the previous paragraphs, both in chronic hepatitis C and in neuroinflammation induced by sepsis and by LPS, as well as experimental models of multiple sclerosis, spinal cord injury, and respiratory diseases. As we previously commented, two preliminary reports have suggested the clinical efficacy of amantadine and memantine in the treatment of COVID-19 [34,35].

This review has as its main limitation the fact that most of the studies published on the anti-inflammatory effects of amantadine and memantine (Table 1 and Table 2) are related to chronic, rather than acute, models of neuroinflammation. Two studies were performed with experimental models of pulmonary disease, and other studies on chronic hepatitis C. Unfortunately, to our knowledge no studies have been published to date on cardiovascular inflammation (vascular inflammation being one of the main pathogenic factors), although the study by Wang et al. [27] reported anti-inflammatory effects of memantine in the human microvascular endothelium. In addition, clinical studies in humans with hepatitis C virus infection were open-label and/or included a low number of patients.

Taking into consideration the previously mentioned limitations, in the light of available evidence, we suggest that the addition of amantadine or memantine as add-on therapy to other COVID-19 therapies could hypothetically improve the outcome. Therefore we propose the development of pilot double-blind studies evaluating the efficacy of amantadine and memantine as add-on therapy in the treatment of COVID-19.

## Figures and Tables

**Figure 1 jpm-10-00217-f001:**
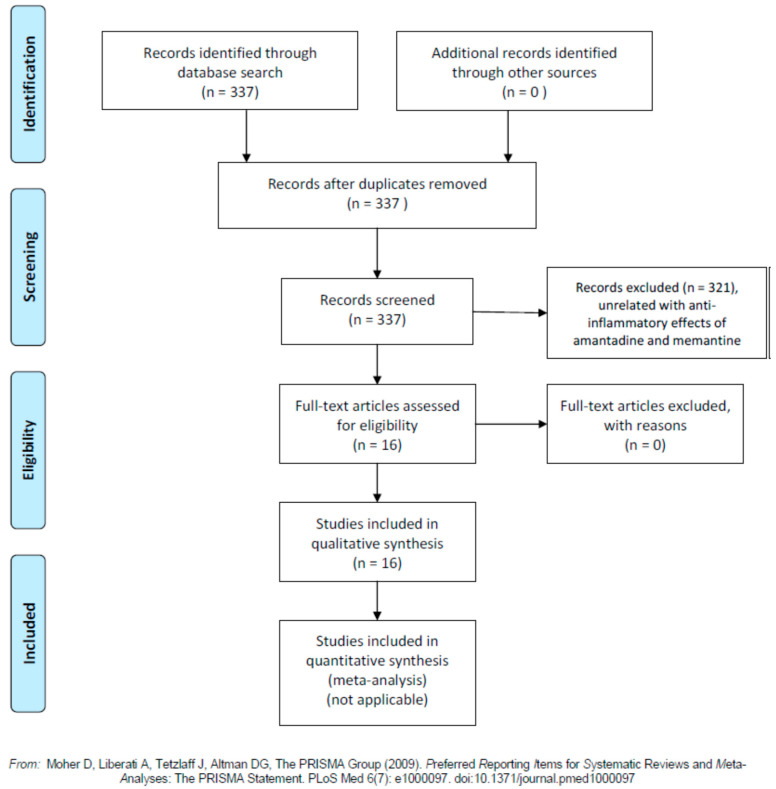
Flowchart of the studies on anti-inflammatory effects of amantadine and memantine. For more information, visit www.prisma-statement.org.

**Figure 2 jpm-10-00217-f002:**
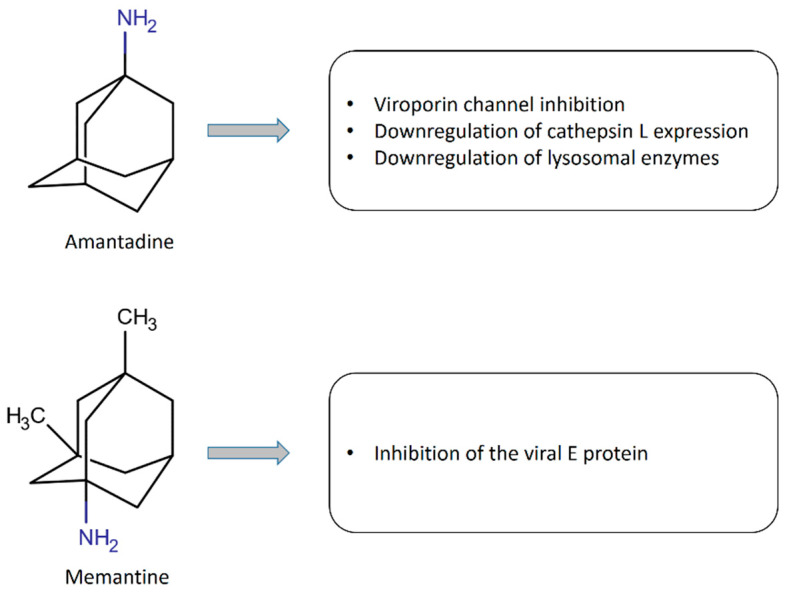
Mechanisms involved in the antiviral effects of amantadine and memantine.

**Table 1 jpm-10-00217-t001:** Studies assessing the anti-inflammatory effects of amantadine.

Authors, Year [Ref]	Country	Model	Study Design	Main Findings
Gelderblom et al., 2007 [14]	The Netherlands	Human hepatitis C	Open-label, single-arm study. 57 patients with initial viral response to IFNα2b but with the reappearance of low-level HCV RNA at weeks 16 to 20. Initial induction therapy with amantadine, IFNα2b, and ribavirin during 6 weeks, and substitution of IFNα2b for pegylated IFNα2b later with a follow-up period of 24–48 weeks	34 patients (59.6%) showed sustained virological response (SVR) and the other 23 (40.4%) showed breakthrough or relapse.
Palabıyıkoğlu et al., 2012 [15]	Turkey	Human hepatitis C	Randomized controlled trial involving 43 patients who did not respond previously to the combination IFNα-2a plus ribavirin for 48 weeks. Treatment with pegylated IFN-α2a (180 μg/kg /week) plus ribavirin (1000–1200 mg/day) and amantadine (200mg/day, 21 patients) or with amantadine alone (200mg/day, 22 patients) for 48 weeks	Significantly higher frequency of SVR in a group of 21 patients treated with pegylated IFN-α2a + ribavirin + amantadine than in the other group of 22 patients treated with amantadine alone.
Younossi et al., 2005 [16]	United States of America	Human hepatitis C	Multi-centre, open-label clinical trial involving 168 patients (69 drug-naïve and 99 non-responders to previous therapy, 32 of them to IFN-α only and 67 to IFN-α plus ribavirin). Both groups started therapy with pegylated IFN-α2b (1.5 μg/kg week), ribavirin (1000–1200 mg/day), and amantadine (200 mg/day), for 4 weeks, followed by pegylated IFN-α2b (0.5 μg/kg week ribavirin (1000–1200 mg/day), and amantadine (200 mg/day), for another 20 weeks. Patients with undetectable HCV RNA at week 24 continued this regimen for a total of 48 weeks and were followed for another 24 weeks. Patients with undetectable virus (<50 IU/mL) after 24 weeks of follow-up were considered to have SVR.	Of the entire cohort, 35 (21%) discontinued early due to side effects or loss to follow-up. SVR and end-of-treatment virological response (ETR) in 47.8% 34.3% and of patients of the drug-naïve group vs in 28.6% and 19.4% for the group who had previously failed to respond to a course of treatment (*p* = 0.01).
Parolin et al., 1999 [17]	Brazil	Human hepatitis C	Open-label study involving 18 patients with hepatitis C non-responders to IFN-α. Treatment with amantadine 200 mg/day.	No significant reduction of serum alanine aminotransferase levels and of viral load between baseline and final values
Kamar et al., 2004 [18]	France	Human hepatitis C	Open pilot study involving 8 hepatitis C virus-positive renal-transplant patients with chronic active hepatitis and increased alanine aminotransferase (ALT) levels. Treatment with amantadine 200 mg/day. Follow-up of 6 months.	Non-significant decrease in hepatitis C viremia, in aspartate aminotransferase activity, and no changes in liver histology. Significant decrease in ALT activity
Yagura & Harada, 2001 [19]	Japan	Human hepatitis C	Open-label prospective study involving 25 drug-naïve and 33 non-responders to IFN-α. Treatment with amantadine 100 mg/day during 4 months and 200 mg/day during the subsequent 2 months.	Reduction of ALT levels in 75% of patients at 4 months and 85% at 6 months after therapy. HCV RNA levels did not modify during the treatment period.
Asai et al., 2001 [20]	Japan	Bronchial epithelial cells cultures	Infection of bronchial epithelial cells in culture with influenza virus and amantadine. Measurement of p38 MAP kinase and JNK activation in the cells and RANTES concentrations in the culture supernatants	Amantadine-induced inhibition of virus replication decreased p38 MAP kinase and JNK activity and expression of RANTES in infected cells. Amantadine did not inhibit p38 MAP kinase and JNK activation induced by tumour necrosis factor α (TNF-α).
Sulkowski et al. 2013 [21]	Poland	Female rats with experimental allergic encephalomyelitis (EAE)	Screening analysis of inflammatory mediators (cytokines and chemokines) in control rats, rats with untreated EAE, and groups rats of EAE treated with amantadine (100 mg/kg b.w./day), memantine (60 mg/kg b.w./day), LY 367385 (10 mg/kg b.w./day, an antagonist of mGluR1) or MPEP (10 mg/kg b.w./day, an mGluR5 antagonist)	Amantadine and memantine suppressed neurological symptoms of disease in EAE rats and reduced expression of pro-inflammatory cytokines in the brain, while antagonists of metabotropic glutamate receptors (mGluR) did not affect the inflammatory process and the neurological symptoms of EAE rats.
Xing et al., 2018 [22]	United States of America and China	Experimental model of sepsis-induced cognitive dysfunction by caecal ligation and puncture (CLP) in male mice	Measurements of learning and memory, and brain cortex expression of several mediators of inflammation in mice controls (not being exposed to surgery or any drugs), mice treated with amantadine, CLP, or CLP plus amantadine 1 (the first dose of amantadine was given intraperitoneally 15 min before the surgery to create CLP), and CLP plus amantadine 2 (the first dose of amantadine was given intraperitoneally 6 h after the creation of CLP).	Amantadine attenuated CLP-induced neuroinflammation in the hippocampus (i.e., interleukin (IL-1β and IL-6 levels) and dysfunction of learning and memory, but did not have significant effects on the expression of toll-like receptors (TLRs)
Dogan & Karaka, 2020 [23]	Turkey	Experimental spinal cord injury in male rats	Spinal cord injury (SCI) induced by removing spinous processes and laminar arcs of T5–12 followed by laminectomy T11-T12 and compression with a clip for 60 s at this level. Measurement of oxidative stress, inflammation, and angiogenesis markers (histological examination of spinal cord sections) in control rats, rats with SCI, and rats with SCI treated with amantadine.	Amantadine had an inhibitory effect of oxidative stress, inflammation, and apoptosis, and ameliorated SCI effects by inducing angiogenesis.
Ossola et al., 2011 [24]	Finland and the United States of America	Primary cultures with different composition of neurons, microglia, and astroglia obtained from the mesencephalon of female rats	Treatment of primary cultures with two dopaminergic neurotoxins: lipopolysaccharides (LPS) and 1-methyl-phenylpyridinim ion (MPP^+^). Measurement of production of tumour necrosis factor-alpha, Prostaglandin E2 (PGE2) release, nitric oxide (NO) production, glial-derived neurotrophic factor (GDNF) gene expression, and changes in intracellular Ca^2+^.	Amantadine protected rat midbrain cultures from MPP+ and LPS toxicity by inhibiting the release of microglial pro-inflammatory factors, and increasing the expression of GDNF in astroglia, while NMDA receptor inhibition was not related to the neuroprotective effect.

**Table 2 jpm-10-00217-t002:** Studies assessing the anti-inflammatory effects of memantine.

Authors, Year [Ref]	Country	Model	Study Design	Main Findings
Sulkowski et al. 2013 [21]	Poland	Female rats with experimental allergic encephalomyelitis (EAE)	Screening analysis of inflammatory mediators (cytokines and chemokines) in control rats, rats with untreated EAE, and groups rats of EAE treated with amantadine (100 mg/kg b.w./day), memantine (60 mg/kg b.w./day), LY 367385 (10 mg/kg b.w./day, an antagonist of mGluR1) or MPEP (10 mg/kg b.w./day, an mGluR5 antagonist)	Amantadine and memantine suppressed neurological symptoms of disease in EAE rats and reduced expression of pro-inflammatory cytokines in the brain, while antagonists of metabotropic glutamate receptors (mGluR) did not affect the inflammatory process and the neurological symptoms of EAE rats.
Rosi et al. 2006 [25]	Italy	Neuroinflammation induced by chronic LPS infusion into the 4th ventricle of rats	Behavioural testing, histological examination, immunofluorescence studies, fluorescence in situ hybridization and confocal microscopy in the dorsal hippocampus in two groups: (LPS-infused; LPS-infused + memantine; and CSF-infused) and caged control groups (LPS-infused; LPS-infused + memantine; and artificial CSF-infused)	Memantine treatment attenuated LPS-induced spatial learning and memory impairments, reduced the number of activated microglia in the hippocampus without affecting resident microglia, and returned Arc (*a*ctivity-*r*egulated *c*ytoskeletal associated protein) expressing neuronal populations to control levels after LPS-infusion
Wei et al., 2016 [26]	China	6-hydroxydopamine (OHDA)-lesioned PC12 cells	4 experimental groups treated with either Dulbecco’s modified Eagle’s medium), memantine (10 μM), 6-OHDA (100 μM), or 6-OHDA (100 μM) plus memantine (10 μM). Quantification of PC12 cell death (via the CCK8 and the apoptotic cells) and measurement of lactic dehydrogenase (LDH), glutamate, IL-6, and TNF-α; immunofluorescence for Nurr1, Nur77, Cyt c, and HSP60; and protein extraction for Nurr1, Nur77, tyrosine hydroxylase (TH), dopamine transporter (DAT), brain-derived neurotrophic factor (BDNF), phosphatidylinositol 3 kinase (PI3K)/p-PI3K, AKT/p-AKT, Cyt c, Lamin B1, and β-actin. Inhibition of the extracellular signal-regulated protein kinases (ERK), c-JunN-terminal kinase (JNK) and p38 mitogen-activated protein kinases (MAPKs)	Memantine incubation prevented the increase of inflammatory mediators (IL-6, TNF-α) and oxidative predictors (glutamate and LDH release), reversed the decrease in the total level of Nurr1, and attenuated in a dose-dependent manner the increased level of Nur77, all induced by 6-OHDA in PC12 cells. Memantine decreased Cyt c and HSP60 release from mitochondria of PC12 cells. Memantine restored the reduced cell viability, attenuated the increased apoptosis, and restored the reduction in the levels of TH and DAT in 6-OHDA-lesioned PC12 cells
Wang et al., 2017 [27]	China	Primary human brain microvascular endothelial (HBMVE) cells cultures	Pre-treatment of HBMVEs with 10 or 20 μM memantine for 12 h (or 24 h) and incubation with 5 ng/mL TNF-α for another 12 h (or 24 h). Measurement of adhesion of Human monocytic leukaemia cell line THP-1 cells.	Pre-treatment with memantine caused an important suppression of TNF-α-induced binding of THP-1 cells to HBMVEs in a dose-dependent manner; rescued TNF-α-induced disruption and interfered with THP-1 cells transmigration across a blood-brain BBB model, and prevented the increased expression of cell adhesion molecules (ICAM-1, VCAM-1, and E-selectin) induced by TNF-α.
Cheng et al., 2019 [28]	China	Chronic obstruction pulmonary disease induced in mice by cigarette-smoking and LPS and Raw264.7 cells cultures	Four experimental groups for mice: controls, cigarette-smoking plus LPS, memantine, and cigarette-smoking plus LPS plus memantine. Determinations in plasma, bronchoalveolar lavage fluid, and lung tissues. Two experimental groups for Raw265.7 for cells: cigarette-smoking exposure plus LPS, and cigarette-smoking exposure plus LPS plus memantine. Measurement of Ca^2+^ influx, glutamate content, cytokines (TNF-α, IL-6, and IFN-γ) and proteins (p-ERK42/44, ERK1/2, p-AKT, NMDAR-1, xCT, GADPH)	Memantine attenuated significantly the increase in the release of cytokines (TNF-α, IL-6, and IFN-γ) and glutamate, and the increase of protein levels of NR-1 and xCT and Ca^2+^ influx, and the activation of the ERK1/2 pathway in the two models
Mishra et al., 2020 [29]	India	Neuroinflammation and memory impairment induced by intracerebroventricular injection of streptozotocin (STZ) in rats.	Experimental groups: controls, STZ, STZ plus ibuprofen (200 μM), and STZ plus memantine (5 µM). Measurement of the expression of NMDAR1, NMDAR2A, NMDAR2B, calcium/calmodulin-dependent protein kinase II subunit α (CaMKIIα), cyclic AMP-response element-binding (CREB) protein, Calpain, and Caspase 3; and levels of glial fibrillary acidic protein (GFAP), TNF-α, inducible nitric oxide synthase (iNOS), and IL-10.	Memantine (and not Ibuprofen) was able to prevent the increase in the expression of NMDAR1, NMDAR2B, Calpain, and Caspase 3 expression, and the decrease in the level of NMDAR2A, CaMKIIα, and CREB protein expression induced by STZ treatment. Memantine and ibuprofen restored the increase in the level of GFAP, TNF-α, and iNOS), and the decrease in the level of IL-10 induced by STZ treatment.

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
