# Peer review of "Anti-Inflammatory Effects of Amantadine and Memantine: Possible Therapeutics for the Treatment of Covid-19?"

_jpm, 2020, doi:10.3390/jpm10040217_

Round 1

Reviewer 1 Report

The authors conducted a review of data regarding the anti-inflammatory effects of amantadine and memantine in inflammation. Based on the observed anti-inflammatory effects shown by these two drugs in chronic hepatitis C, in neuroinflammation induced by sepsis and by lipopolysaccharides, as well as in experimental models of multiple sclerosis, spinal cord injury, and respiratory diseases, the authors propose the need to investigate their potential utility in COVID-19. 

While at present there are not sufficient data supporting the potential efficacy of these drugs, the hypothesis is interesting and the topic is obviously of high interest. 

The review is short and easy to read but might benefit of some changes in order to improve clarity and its potential utility for a reader. 

While the authors report a search strategy, the review can be considered as a descriptive rather than systematic review, as for instance number of studies excluded according to defined exclusion criteria are not reported. Nonetheless, I still think it would be of help to specify which criteria the authors followed in order to select included studies (for instance specify whether all studies either preclinical, conducted in cell lines derived from patients, commercial cell lines and so on were included).

Sections 2 and 3, related to the anti-inflammatory effects of amantadine and memantine, are at present mostly a list of studies. I think these sections would really benefit from a more extensive presentation of results, including their strenghts and most importantly limitations of presented studies. At present, it is not easy to distinguish which results the authors believe to be more or less relevant so it would be useful to present the studies from a more critical point of view. 

At page 2, line 46, please replace " because the development..." with "because  ofthe development..."

At page 2, line 66, this sentence seems incomplete "... references strictly to with this issue were selected."

At page 4, line 156:

- the following statement should be updated in light of new data (if available): "According to Kumar et al. [1] up to 29th April 2020, no drug has been approved by the United States Food and Drug Administration that can prevent COVID-19". 

- also, the cited reference does not seem to be the correct one.

Finally, I suggest to present a more comprehensive view of available studies investigating the effect of hydroxychloroquine in COVID-19 rather than just results from a study. 

Author Response

The authors conducted a review of data regarding the anti-inflammatory effects of amantadine and memantine in inflammation. Based on the observed anti-inflammatory effects shown by these two drugs in chronic hepatitis C, in neuroinflammation induced by sepsis and by lipopolysaccharides, as well as in experimental models of multiple sclerosis, spinal cord injury, and respiratory diseases, the authors propose the need to investigate their potential utility in COVID-19. While at present there are not sufficient data supporting the potential efficacy of these drugs, the hypothesis is interesting and the topic is obviously of high interest. The review is short and easy to read but might benefit of some changes in order to improve clarity and its potential utility for a reader. 

  • While the authors report a search strategy, the review can be considered as a descriptive rather than systematic review, as for instance number of studies excluded according to defined exclusion criteria are not reported. Nonetheless, I still think it would be of help to specify which criteria the authors followed in order to select included studies (for instance specify whether all studies either preclinical, conducted in cell lines derived from patients, commercial cell lines and so on were included). OK, we have added a figure with the flowchart for the selection of studies and 2 tables with details of the studies related to the anti-inflammatory effects of amantadine and memantine. We also have added a new study published since the first submission of this paper.
  • Sections 2 and 3, related to the anti-inflammatory effects of amantadine and memantine, are at present mostly a list of studies. I think these sections would really benefit from a more extensive presentation of results, including their strenghts and most importantly limitations of presented studies. At present, it is not easy to distinguish which results the authors believe to be more or less relevant so it would be useful to present the studies from a more critical point of view.  As we have mentioned in the previous point, we have added 2 tables with details of the studies related to the anti-inflammatory effects of amantadine and memantine. We also have added a new study published since the first submission of this paper. Limitations of these studies have been added in the discussion section.
  • At page 2, line 46, please replace " because the development..." with "because  ofthe development..." OK, done
  • At page 2, line 66, this sentence seems incomplete "... references strictly to with this issue were selected." OK, corrected
  • At page 4, line 156: - the following statement should be updated in light of new data (if available): "According to Kumar et al. [1] up to 29th April 2020, no drug has been approved by the United States Food and Drug Administration that can prevent COVID-19".  OK, this sentence was deleted and the statement updated. - also, the cited reference does not seem to be the correct one. The reference regarding that paragraph was deleted. 
  • Finally, I suggest to present a more comprehensive view of available studies investigating the effect of hydroxychloroquine in COVID-19 rather than just results from a study. OK, we have added a comment regarding the lack of recommendation of these drugs in the US and by the WHO in the discussion.

Reviewer 2 Report

The authors propose to treat COVID-19 patients using AMANTADINE AND MEMANTINE in this short communication.

Major concerns:

1. Up-to-date, no reports support the use of amantadine and memantine in COVID-19 patients. Though the medicines have been used in viral infection including influenza A infection, it does not necessarily support that the medicines are effective in treatment of COVID-19. The study on old drug for new use are encouraged, stringent clinical trails could be designed to test the hypothesis. Therefore, this manuscript is premature, results from a small scale clinical trail will strongly backup the propose. 

2. The molecular basis of the medicines should be discussed in detail, a graphic summary is helpful for readers to understand their anti-inflammation effects.  

3. In addition, the application of these medicines in a range of distinct clinical conditions should be carefully discussed.    

Author Response

The authors propose to treat COVID-19 patients using AMANTADINE AND MEMANTINE in this short communication.Major concerns:

  • Up-to-date, no reports support the use of amantadine and memantine in COVID-19 patients. Though the medicines have been used in viral infection including influenza A infection, it does not necessarily support that the medicines are effective in treatment of COVID-19. The study on old drug for new use are encouraged, stringent clinical trails could be designed to test the hypothesis. Therefore, this manuscript is premature, results from a small scale clinical trail will strongly backup the propose.  We have added preliminary data on COVID 19 patients treated with amantadine.
  • The molecular basis of the medicines should be discussed in detail, a graphic summary is helpful for readers to understand their anti-inflammation effects.  We have included a graphic summary (Figure 2) indicating the mechanisms involved in the antiviral effects of both drugs.
  • In addition, the application of these medicines in a range of distinct clinical conditions should be carefully discussed.  Done in the introduction section

Reviewer 3 Report

The authors have done a literature search on the anti-inflammatory effects of Amantidine family drugs and based upon the literature suggest that the drugs might be helpful for patients with Covid19.  While the drugs appear to have the desired effects, the literature used i mostly unrelated to viral infections which might be relevant.  Nevertheless the literature supports the conclusions of the authors.  I would suggest that the discussion be modified however.  The authors state that no therapeutics are presently available for treating Covid19 and then discuss the use of hydroxychloroquine and azithromycin.  I believe that this should be modified for two reasons.  First, there are now two drugs recommended for treatment at least in the US, dexamethasone and remdesivir.  Both have now been approved specifically for treating Covid19.  In addition, recombinant monoclonal antibodies have also been granted special use and have shown promise.  In contrast, at least in the US and by WHO the combination of hydroxychloroquine and azithromycin and no longer recommended for treatment.  This should be noted.  

Author Response

The authors have done a literature search on the anti-inflammatory effects of Amantidine family drugs and based upon the literature suggest that the drugs might be helpful for patients with Covid19.  While the drugs appear to have the desired effects, the literature used i mostly unrelated to viral infections which might be relevant.  Nevertheless the literature supports the conclusions of the authors.  I would suggest that the discussion be modified however. 

  • The authors state that no therapeutics are presently available for treating Covid19 and then discuss the use of hydroxychloroquine and azithromycin.  I believe that this should be modified for two reasons.  First, there are now two drugs recommended for treatment at least in the US, dexamethasone and remdesivir.  Both have now been approved specifically for treating Covid19.  OK, we have added a comment on dexamethasone and remdesivir and other emergent therapies in the discussion.
  • In addition, recombinant monoclonal antibodies have also been granted special use and have shown promise.  OK, added in the discussion
  • In contrast, at least in the US and by WHO the combination of hydroxychloroquine and azithromycin and no longer recommended for treatment.  This should be noted.  OK, this has been noted

Round 2

Reviewer 1 Report

The authors addressed all comments and also updated the review. I have no further comments. 

Reviewer 2 Report

I do not have further concerns.